# Shape Evolution in Two Acts: Morphological Diversity of Larval and Adult Neoaustraranan Frogs

**DOI:** 10.3390/ani14101406

**Published:** 2024-05-08

**Authors:** Diego Almeida-Silva, Florencia Vera Candioti

**Affiliations:** 1Unidad Ejecutora Lillo, Consejo Nacional de Investigaciones Científicas y Técnicas–Fundación Miguel Lillo, San Miguel de Tucumán 4000, Argentina; florivc@gmail.com; 2Centro de Ciências Naturais e Humanas, Universidade Federal do ABC, São Bernardo do Campo 09606-045, SP, Brazil

**Keywords:** adaptive decoupling hypothesis, evolutionary convergence, shape disparity, complex life cycles, tadpoles, Alsodidae, Batrachylidae, Cycloramphidae, Hylodidae

## Abstract

**Simple Summary:**

Understanding how traits evolve across different life stages is crucial for species with complex life cycles. We investigated how tadpole morphology relates to adult shape in Neoaustrarana frogs, finding that larval and adult shapes show low correlation. Our study, encompassing 83 species across four families, highlights the importance of considering ontogenetic stages in evolutionary research. We observed distinct patterns among families, with larval morphospace exhibiting higher phylogenetic structure than adult morphospace. Our results support the Adaptive Decoupling Hypothesis as a mechanism driving phenotypic diversity, shedding light on the early evolution of Neoaustrarana frogs and emphasizing the need for further research in this area.

**Abstract:**

Phenotypic traits can evolve independently at different stages of ontogeny, optimizing adaptation to distinct ecological contexts and increasing morphological diversity in species with complex life cycles. Given the relative independence resulting from the profound changes induced by metamorphosis, niche occupation and resource utilization in tadpoles may prompt evolutionary responses that do not necessarily affect the adults. Consequently, diversity patterns observed in the larval shape may not necessarily correspond to those found in the adult shape for the same species, a premise that can be tested through the Adaptive Decoupling Hypothesis (ADH). Herein, we investigate the ADH for larval and adult shape differentiation in Neoaustrarana frogs. Neoaustrarana frogs, particularly within the Cycloramphidae family, exhibit remarkable diversity in tadpole morphology, making them an ideal model for studying adaptive decoupling. By analyzing 83 representative species across four families (Alsodidae, Batrachylidae, Cycloramphidae, and Hylodidae), we generate a morphological dataset for both larval and adult forms. We found a low correlation between larval and adult shapes, species with a highly distinct larval shape having relatively similar shape when adults. Larval morphological disparity is not a good predictor for adult morphological disparity within the group, with distinct patterns observed among families. Differences between families are notable in other aspects as well, such as the role of allometric components influencing shape and morphospace occupancy. The larval shape has higher phylogenetic structure than the adult. Evolutionary convergence emerges as a mechanism of diversification for both larval and adult shapes in the early evolution of neoaustraranans, with shape disparity of tadpoles reaching stable levels since the Oligocene. The widest occupation in morphospace involves families associated with dynamically changing environments over geological time. Our findings support the ADH driving phenotypic diversity in Neoaustrarana, underscoring the importance of considering ontogenetic stages in evolutionary studies.

## 1. Introduction

The interpretation of shape as a phenotypic trait subject to variation throughout ontogeny is one of the central issues in developmental biology [1,2,3]. Many species of vertebrates differ conspicuously from each other in shape but exhibit considerable morphological similarity throughout embryonic development, an observation known for centuries, e.g., [4,5]. This phenotypic differentiation becomes particularly evident for species with complex life cycles [6,7], such as anuran amphibians. In most anurans, ontogeny is divided into two discrete phases, with distinct ecological characteristics and well-defined body plans, separated by a process of metamorphosis. Ecologically, the shift from a predominantly aquatic to primarily terrestrial niche entails essential differences in resource utilization (feeding, microhabitat). This may represent a turning point in phenotypic diversification patterns before and after metamorphosis [8,9] and may be interpreted in the conceptual framework of the hypothesis of adaptive decoupling (ADH). The ADH proposes the independent evolution of phenotypic traits expressed at different stages of the life history, many of which lack equivalents in other ontogenetic phases [10]. Its implication would be an optimization of adaptation in both ecological contexts [8,9]. While expected in animals with complex life cycles, the discontinuity between pre- and postmetamorphic traits should be verified for each trait of interest, as it is not obligatory, e.g., [10,11,12,13,14]. In some cases, the association between traits across ontogenetic phases may confer adaptive value per se; in others, the genetic factors acting in all phases may be interdependent [15,16,17].

Given that a central prediction arising from ADH is the expectation of greater morphological and ecological diversification in response to dissociation [18], groups with high ecomorphological diversity may be particularly suitable for adaptive decoupling. In this sense, the anuran family Cycloramphidae stands out as a remarkable group, comprising species whose tadpoles greatly differ from pond-type tadpoles characteristic of other groups. The family currently comprises 37 species organized into 2 genera, *Thoropa* and *Cycloramphus* [19], primarily distributed in the Brazilian Atlantic Forest. These frogs are typical inhabitants of forested areas, with species associated with leaf litter, rocky outcrops, or swift streams [20,21,22,23,24]. Adult specimens breed in aquatic or terrestrial environments [20,22,25], five of them having semifossorial habits (gr. *C. bolitoglossus*) [20,22,26,27]. In turn, tadpoles of most species are semiterrestrial, dwellers of wet rocks and rivulets, and characterized by a distinct, streamlined body shape, e.g., [22,23,24,25,26]. Ten species exhibit endotrophic nidicolous development, whereby a free-living, non-feeding larval stage occurs coupled with some anatomical modifications [28].

Recent phylogenetic analyses consistently recover Cycloramphidae along with another group from the Atlantic Forest (Hylodidae) and two Patagonian families (Alsodidae and Batrachylidae), in a well-supported clade called Neoaustrarana [29,30,31,32,33]. Despite tadpole morphology in these other families being somewhat closer to the typical larval form of other anurans, ecological niche usage denotes an interesting diversity, allowing for a comparative approach within a broader diversification context. Within these lineages, species occur in lotic (e.g., species of Hylodidae) [34,35,36], lotic–lentic, and lentic environments (e.g., species of Alsodidae and most Batrachylidae) [37,38,39,40,41], with some species exhibiting terrestrial oviposition (*Batrachyla* spp.) [37,40,42] and endotrophic nidicolous development (*Eupsophus* spp.) [40,43,44]. An example of this diversity is shown in Figure 1.

In this work, we follow the pioneer approach by [45], who compared larval and adult shape disparity in Australian frogs and discussed patterns of convergent and divergent evolution associated to morphological diversification along ontogeny. Departing from Cycloramphidae and its salient tadpole ecomorphological diversity, we explore larval and postmetamorphic shape diversification in the broader context of Neoaustrarana. We inquire about the influence of phylogenetic and functional diversity on shape variation and reconstruct morphological evolution along evolutionary time. Finally, we discuss our results in the context of the adaptive decoupling hypothesis.

## 2. Materials and Methods

### 2.1. Sample and Data Acquisition

To analyze the relationship between larval and adult morphologies, we restricted our sample to species for which data on both developmental stages are available. Our study focuses on 83 out of 127 species described in four neoaustraranan families [30]: 22/30 known species of Alsodidae, 8/12 species of Batrachylidae, 20/37 species of Cycloramphidae, and 33/48 species of Hylodidae. We compiled a dataset with external morphology description per species, with information taken from photographs on vouchered biological material, and illustrations and data available in the literature (Appendix A). For larval morphology, we followed a two-dimensional geometric morphometric approach. On a set of 112 photographs/drawings of tadpoles (*n* = 1 or 2 per species; developmental stages 26–41 [46]), a set of 32 landmarks was defined based on [47] (Appendix A). Landmarks were digitized by the same person, and configurations were digitally unbent to correct tail curvature due to fixation artifacts [48]. TpsDig 2.17 and TpsUtil 1.58 were used [48,49], and mean shape values per species were generated with the geomorph R package [50]. For adult morphology, we took linear measurements from dorsal and lateral images of mostly male specimens (4 females and 11 individuals with no sex information; Appendix A). Ten measurements were selected following [51] (Appendix A) and registered by the same person using ImageJ [52]. For species where it was not possible to obtain biological material or illustrations, we used measurements available in original descriptions, taxonomic revisions, or general literature. We used the missMDA r package [53] to impute missing values based on principal component analysis (PCA). To explore preliminarily ecomorphological diversity, we assigned species to larval and adult functional groups, following [54] and species accounts (Appendix A).

### 2.2. Statistical Analysis

All the following procedures and analyses were performed in the R statistical environment v.4.3.2 [55]. For the larval dataset, we derived a shape coordinate matrix through a Generalized Procrustes Analysis (GPA) on mean shape data. To control for the effect of larval developmental stage in shape variation, we performed a Procrustes ANOVA with genus, family, ecomorphological guild, and developmental stage as factors. Gosner developmental stages were collapsed into three categories as follows: limb bud (Stages 26–30), autopodium development (Stages 31–37), and tarsal tubercle differentiation (Stages 38–41) [46], and *R^2^* and *p*-values were considered to assess factor relative importance [50]. Other variables known to have a significant effect on tadpole shape, especially at an intraspecific level (e.g., competition, predation) [56,57], were not considered. For the adult dataset, we transformed measurements (excepting snout–vent length) through the log-shape ratios approach (LSR), a natural logarithm transformation of data corrected by geometric mean [58]. Both GPA and LSR approaches are analogous regarding the correction for size while retaining the allometric shape variation [45]. First, to explore shape variation along ontogeny, we generated larval and adult morphospaces by performing a Principal Component Analysis (PCA) on shape matrices (larval Procrustes coordinates and adult size-corrected measurements) through the R package stats [55]. To investigate allometry by family, we employed multivariate regressions of shape on size (larval total length and adult snout–vent length as natural logarithm-transformed values) on the whole dataset and per family. Allometric trends were illustrated in geomorph, by summarizing shape as the first principal component of a matrix of predicted shapes as well as the distribution of raw regression scores (projections of data points onto an axis in the direction of the regression vector) [50,59]. The statistical significance of regressions was evaluated by permutations (9999 iterations).

To consider the role of evolutionary history in morphological diversification across neoaustraranan larval and adult morphospaces, we employed the dated topology recently proposed by [31]. Most of the species comprising our sample are represented in this phylogenetic hypothesis (74 out of the 83 species we sampled). Utilizing the PC1 scores per species, we used the geomorph R package to compute the phylogenetic signal of shape for both datasets using Blomberg’s K. These K values tend to be higher than 1 when closely related species exhibit more similar shapes than expected, while K values less than 1 indicate lower similarity among closely related species. We projected the phylogeny into morphospaces, and we used the convevol R package [60] to perform Stayton’s C1–C4 measures of convergent evolution, as indicated by branch projections into the same region of the phylomorphospace [61]. We assessed the alignment of species positioning on morphospaces by using PC1 scores. Stayton’s C1–C4 measures assess the reduction in distance between taxa over time, indicating increasing morphological similarity (C1), the magnitude of morphological change between convergent taxa (C2), and a normalization of this magnitude relative to the evolutionary time of the whole phylogenetic tree (C3) or a specific clade (C4). Together, these measures provide a framework to infer the correlation between phenotypes according to phylogenetic relationships and functional groups. By considering both the magnitude of changes and the evolutionary context, they offer distinct insights into larval and adult morphological convergence. We evaluated the significance of convergences through permutations (1000 iterations) under the reference phylogenetic hypothesis. To address uncertainty in tree topology and divergence time estimates, we used 100 suboptimal trees to repeat the calculation of Stayton’s C1–C4 measures, showing results as boxplots.

To visualize the scale and directionality of larval and adult shape evolution over time, we estimated and plotted ancestral body shape as a traitgram, using phytools [62]. We estimated ancestral shapes by the reconstruction of PC1 scores under maximum likelihood as a continuous character. From the ancestral estimation of morphospace position, we can infer how the morphospace occupation of a given taxa varies along clade diversification. We employed ancestral estimation along phylogenetic branches to examine morphological disparity over time in Neoaustrarana, following the approach suggested by [63]. We defined time bins based on intervals of geological ages along Neoaustrarana node depth, interpolating PC1 scores on the time-scaled tree branches, as in a gradual model [64]. To interpolate data, we used the observed PC1 scores at terminals and the ultra-fast maximum likelihood ancestral state reconstruction for the nodes obtained with the Rphylopars package [65]. Then, we calculated the sum of variances using branch-interpolated values at the midpoint of each time bin to access the morphological disparity through time. The process is analogous to the morphological disparity inferred in geomorph, which is based on Procrustes variance [50].

Considering that adaptive decoupling would result in a break between variation in larval and adult morphological patterns, it is reasonable to hypothesize that this proposition can be assessed by examining the degree of correlation in shape between the two developmental stages. We evaluated the strength of correlation between the two datasets through a Mantel test, using the vegan package [53]. We computed Euclidean dissimilarities between species on tadpole and adult shape matrices and assessed statistical significance of Spearman’s correlation through permutations (9999 iterations). Additionally, we investigated whether shape variation in tadpoles is correlated to adult shape variation. We assessed morphological disparity on tadpole and adult shape matrices and followed [66] to account for uncertainty using a permutation approach. We used the geomorph package to calculate the morphological disparity of subsets of half the total sample, iterating this process along 1000 times. As the general pattern for Neoaustrarana may not adequately describe family-specific trends, we repeated the procedure by family. We calculated the Spearman’s correlation between larval and adult morphological disparities for all cases.

## 3. Results

### 3.1. Shape and Size

Larval and adult morphospaces are shown in Figure 2. The first two principal components explain 87.7% of larval shape variation (Figure 2a). Variation in PC1 is mostly associated with body and tail height increase, with cycloramphid tadpoles clearly diverging from the rest. Distribution along PC2 is explained mainly by body/tail ratio, with some relevant divergences at the genus level (Appendix A). Among the Alsodidae, *Eupsophus* species exhibit longer tails than *Alsodes*; the same trend is observed in *Hylodes* as compared to other hylodids. *Cycloramphus* larvae display a highly variable body/tail ratio, whereas this trait tends to be more consistent among *Thoropa* tadpoles. Procrustes ANOVA highlights the significant effects of family (*R*^2^ = 0.22), genus (*R*^2^ = 0.45), and ecomorphological guild (*R*^2^ = 0.19), on larval shape variation; the effect of developmental stage resulted non-significant (*R*^2^ = 0.008; *p* > 0.6), thus we dismissed this for further interpretation (Appendix A). In adult frogs, the first two principal components explain 64.4% of shape variation (Figure 2b). Variation in PC1 is mostly explained by eye–nostril and internarial distances, with hylodid species tending to show separate nostrils closer to the eyes in comparison with other groups. Increasing scores on PC2 associate mainly with larger interorbital distances and short feet (Appendix A). At the genus level, *Thoropa* frogs tend to have larger feet and shorter interorbital distance than *Cycloramphus*, as seen in *Crossodactylus* regarding other hylodids (Appendix A).

Size variation relates to shape variation in tadpoles stronger than in adults (20.4% versus 7.2%; *p* < 0.001 for both regressions; Table 1). Although a large part of the differences among families appears to be independent from size (57.8% and 19.8% in tadpoles and adults, respectively; *p* < 0.001 for both regressions; Table 1), a significant size–family interaction in the tadpole dataset indicates different allometric patterns at this stage (3.8%, *p* < 0.001). A significant effect of size on shape change is recovered in tadpoles of all families excepting Batrachylidae (Appendix A). Allometric trends highlight how larval allometric patterns in cycloramphids diverge from those of other families, whereas in the adult dataset, overlapping distributions obscure differentiation (Figure 3 and Appendix A). The high shape–size variation rate in larval cycloramphids implies a rapid change to flattened tadpoles as size increases (12.3% of shape change explained by size increase; *p* = 0.04). Hylodid tadpoles reach the largest sizes, and a comparatively small but still significant effect of size on shape change is recovered (6.7%; *p* = 0.03). In adults, allometry is recovered significantly in Batrachylidae and Hylodidae (Appendix A), with different shape/size variation patterns apparently mostly related to interorbital distance and foot length (Figure 3 and Appendix A).

### 3.2. Morphological Evolution

Phylogenetic structure of shape variation is strong in neoaustraranan tadpoles (K = 4.26; *p* = 0.001). Phylomorphospace shows a distinct distribution of cycloramphids, and within this family closely related species are also morphologically similar (Figure 4a). Endotrophic nidicolous tadpoles distribute along the region of morphospace intermediate to tadpoles of other families. Stream-dweller tadpoles of Alsodidae and Hylodidae vary widely in shape, and endotrophic tadpoles of alsodid *Eupsophus* appear scattered among mostly lotic hylodids. In cycloramphids and hylodids, a significant part of shape similarity is due to morphological convergence (C1 values > 40%; Table 2). Taking into account the evolutionary time of cycloramphid and hylodid lineages, C3 values suggest that about a quarter of the morphological change in tadpoles can be attributed to convergence throughout evolutionary time. Although convergence is identifiable, C2 values indicate that the magnitude of change was relatively low. Values regarding ecomorphological groups show similar trends, excepting endotrophic tadpoles where no significant morphological convergence is revealed. All these findings remain consistent whether Stayton’s measures of convergent evolution are applied, regardless of addressing uncertainties (Appendix A) in tree topology and divergence time estimates or not.

Instead, shape variation is not phylogenetically structured in adult neoaustraranans (K = 0.44; *p* = 0.001). Only in Hylodidae, convergence appears to have influenced significantly on adult morphological evolution (Table 2 and Figure 4b). Approximately 40% of shape variation converged in morphospace, as suggested by C1. In turn, the C3 value indicates that shape convergence also played a pivotal role throughout the evolutionary time. Furthermore, the C2 value suggests a shape change of approximately 20% in magnitude from ancestors to extant forms in Hylodidae after evolutionary convergence. Regarding ecomorphological groups, only the torrential group exhibits a significant part of shape convergence (C1 ca. 40% and overall values of C2 and C3 slightly higher than in other clades; Table 2).

Traitgram unveils the occupation of two disparate regions in the larval phylomorphospace, both arising from evolutionary convergence processes occurring from the Middle Eocene through the Oligocene (40–25 mya, approximately; Figure 5a). The diversification of shape in Neoaustrarana during the Miocene leads to an expansion in morphological variation within the clade, notably pronounced in Cycloramphidae compared to the other families. This increase in shape diversity along the Miocene is reinforced by a peak in morphological disparity early in this period, which has remained relatively constant so far. Conversely, the traitgram for adult stages suggests significant internal morphological diversification within each family from their ancestors (Figure 5b). This outcome is particularly evident in Hylodidae, where phylogenetically close species underwent evolutionary convergence processes towards specific regions of the phylomorphospace, between the late Oligocene and middle Miocene (25–13 mya, approximately). The morphological disparity among adult neoaustraranans appears to continue increasing over time, reaching greater levels than larval forms, with intensification since the late Miocene.

### 3.3. Adaptive Decoupling Hypothesis

The Mantel test suggests a low correlation between shape variation in larval and adult forms (Mantel statistic r = 0.16; significance: 0.003). A kernel density map (Figure 6a) illustrates the simulated scenario where most neoaustraranans exhibiting low dissimilarity as tadpoles tend to correlate to low dissimilarity as adults as well, whereas those species with high larval dissimilarity do not necessarily lead to a significant increase in dissimilarity during the adult stage. The fact that these two species groups are highly segregated indicates that there is no overall pattern between shape similarity in larval and adult forms. Also, morphological disparity in tadpoles is not correlated to the adult morphological disparity in general terms (r = −0.01; Figure 6b), but analyses per family reveal different patterns. High larval disparity tends to correlate with low adult disparity in Alsodidae (r = −0.4) and Cycloramphidae (r = −0.15), whereas increasing larval disparity leads to an expected increase in adult shape diversity in Batrachylidae (r = 0.75) and Hylodidae (r = 0.42).

## 4. Discussion

In this study, we sought to explore the evolution of shape in Neoaustrarana anurans across their two developmental phases, as well as their interdependence. Overall, our results indicate that larval and adult shapes differ in both morphospace occupation and morphological disparity and are influenced differently by size. Larval and adult shapes also have distinct relationships with phylogeny, reflecting disparate evolutionary trends and histories. In the paragraphs below, we discuss patterns of morphological evolution prior and after metamorphosis at Neoaustrarana level and compare with studies that applied similar approaches on other taxonomic groups. Since in our case the lack of correlation between pre- and postmetamorphic ontogeny at a macroevolutionary level seems to be related with different patterns among families, we also discuss some aspects of intrafamilial diversification.

Previous studies explored larval morphospace at different taxonomic levels, finding low phylogenetic structure and a prevalence of convergence patterns related to microhabitat and behavior. At least four studies, from the intrageneric to suprafamilial scale, reveal a morphological continuum between lentic and lotic species among the main dimensions of shape variation [45,67,68,69,70]. Morphological transformations along these axes often involve external features related to hydrodynamic aspects of living in stagnant vs. flowing water, such as body height, tail length and height, and oral apparatus position. Conversely, larval morphospace is largely structured by phylogeny in Neotropical neoaustraranans. As expected, cycloramphid tadpoles clearly diverge from the rest, with distinct features such as the deeply depressed body and the long tail with low fins. A set of synapomorphies from musculoskeletal system has been proposed additionally to define the family [71]. This divergence apparently took place during the Miocene, and subsequent diversification within the family maintained in part similar morphospace occupancy regarding the ancestor. Although a strong ecomorphological component (see [72]) can be interpreted in the evolution of the clade, the role of common ancestry and phylogenetic inertia is evident in that even endotrophic tadpoles share the extremely attenuated shape of semiterrestrial tadpoles seemingly plesiomorphic for the family ([25]; DAS unpubl. data). In fact, reduction in typically larval features, like that reported in several endotrophic lineages, e.g., [43,44,73,74,75], concerns individual characters (e.g., spiracle and oral disc morphology and keratinization) that do not affect the overall streamlined body shape, e.g., [28,76]. As hinted by the high rate of shape/size variation, significant static allometry appears to be related to ecomorphological diversity in these tadpoles, with a trend of endotrophic tadpoles being smaller and slightly less flattened than semiterrestrial larvae.

In the large cluster formed of the remaining neoaustraranans, Alsodidae and Hylodidae share wide, mostly overlapping distributions in larval morphospace, whereas tadpoles of Batrachylidae show comparatively less shape variation. Hylodids share some similar patterns with Cycloramphidae, in terms of relative occupation of morphospace along evolutionary time. However, shape diversification in this family does not deviate significantly from the pond-type tadpole bauplan. Tadpoles in these three families inhabit different types of water bodies, from ponds and lakes (e.g., *Atelognathus*) [38] to fast-torrent streams (e.g., *Phantasmarana*) [77], but they mostly exhibit benthic behavior and avoid strong currents, being commonly found in slow backwaters, on the bottom of water bodies, or among rocks, e.g., [34,78,79,80,81]. Variations in body and tail height and length appear to be intrageneric changes not strictly related to development in lentic vs. lotic environments. In this context, the distribution of endotrophic nidicolous *Eupsophus* among certain stream-dwelling tadpoles, showing some resemblance in body/tail ratio and fin height as also recovered by [68] in Australian larvae, is at least intriguing. Regarding allometric patterns, size increase appears to have a weak influence on shape variation. This is particularly interesting for hylodids, whose giant tadpoles (e.g., *Phantasmarana* and *Megaelosia* species) appear to maintain similar shapes to those of related tadpoles less than half their length.

Unlike in larval stages, phylogeny alone did not play an important role in the evolution of the adult shape of neoaustraranans. Similarly, functional groups, at least as defined by habitat and microhabitat occupation, in general did not contribute significantly to shape variation. Terrestrial, saxicolous, and aquatic frogs share similar patterns of shape diversity. The congruence in significant morphological convergence both for Hylodidae and the torrential functional group highlights the ecomorphological component involved in the evolution of this family (although a minority of alsodids were coded as torrential, most species in this group are in fact hylodids), along with a possible lack of resolution to define and categorize these frogs following ecological criterions. Despite being strongly associated with streams in the Brazilian Atlantic Forest, hylodids exhibit diversity in their utilization of lotic environments [36]. The definition of “lotic” encompasses highly diverse environments in the Atlantic Forest mountains, including streams of different orders and with great variation in slope. Hence, the distribution of Hylodidae spans from waterfalls [82] (Nascimento et al., 2001) to slow-flowing first-order streams [81,83], encompassing semi-deciduous areas and rocky fields [82,84,85]. The loss of resolution in defining functional groups due to the complexity of the lotic environment also affects other stream-dwelling groups in Neoaustrarana, such as alsodids [39] and cycloramphids [20,24,86]. Morphologically, adult hylodids vary in relation to snout shape in our sample. Many species exhibit separated nostrils close to the eyes, coupled with a well-defined loreal region and a sharp canthus rostralis; these traits are also commonly cited in the taxonomy of the family, e.g., [34,35,87]. Other torrential frogs from unrelated clades typically exhibit these features, e.g., [88,89,90].

As outlined above, profound differences between morphological evolution in larval and adult stages of neoaustraranan frogs point towards the acceptance of the adaptive decoupling hypothesis in the group. An increasing number of studies on this topic have been conducted focusing on anurans, and results stress that differences appear among taxonomic levels and groups, and among characters considered. For example, it is suggested that there is a correlation between larval body size and adult snout–vent length, such that the allometric change in one phase is coupled with the alteration in size of the other [13], at least for some families [91]. At the same time, the rates of phenotypic evolution appear to be decoupled between the two developmental stages, such that an acceleration in the trend of larval size change does not imply a similar process during the adult phase [13]. The rates of phenotypic evolution may also be decoupled regarding the emergence of discrete morphological characters, a process that is evolutionarily much faster in the larval than in the adult phase [92]. As in Neoaustrarana, size-independent shape features in tadpoles and adults are found to be differently influenced by phylogenetic and ecomorphological diversity in previous studies at the intrageneric and macroevolutionary scales [45,69].

The evolutionary history of shape changes in adult Neoaustrarana frogs also contrasts with the premetamorphic pattern. The whole clade underwent significant and constant morphological diversification since the Oligocene, and, especially in Hylodidae, recent changes expanded the occupation of adult morphospace during the late Miocene. It is possible to hypothesize that the evolution of shape may have occurred in response to the diversification occupying very distinct environments. Interestingly, Cycloramphidae and Hylodidae, with the most pronounced morphological disparity, greater exploration in morphospace, and subject to processes of evolutionary convergence, are distributed in association with the mountains of the Brazilian Atlantic Forest. This area underwent deep geomorphological changes over the past 40 million years, including significant orogenic process and compartmentalization of the Serra do Mar and Mantiqueira mountain ranges from the early Oligocene to Miocene [93,94,95,96,97], the introgression of the Paranaense Sea [98], and interglacial events during the late Miocene [99,100]. Conversely, although important changes in temperature and vegetation took place in Patagonia during the Oligocene/Miocene transition [101,102,103], the main changes in relief of the region took place even before, along the Eocene, e.g., [104,105]. Patagonian climate and landscape remained more stable during the Miocene, likely resulting in a more homogeneous environment over the temporal range, which encompasses the emergence of alsodids and batrachylids. The onset of morphological diversification during the ontogeny of Cycloramphidae and Hylodidae, recovered to have occurred earlier than in Patagonian families, could be related to environmental pressures in a highly dynamic landscape.

## 5. Conclusions

Our study on the evolution of shape in Neoaustrarana anurans across larval and adult stages reveals profound differences in morphospace occupation, morphological disparities, allometric patterns, and phylogenetic influences, supporting adaptive decoupling hypothesis. Larval shapes are largely structured by phylogeny, with cycloramphid tadpoles diverging distinctly from other families, while adult shapes show less phylogenetic structure. The exploration of torrential habitats among adult members of the group appears to be strongly associated with shape evolution, despite the challenge of creating categories that reflect the use of lotic environments. The evolutionary history of shape changes suggests a response to the diversification of Neoaustrarana in diverse environments, with pronounced morphological diversification observed in families associated with the dynamic landscape of the Brazilian Atlantic Forest. Overall, our study contributes to a better understanding of how complex interactions between developmental stages, ecological niches, and evolutionary history shape the morphological diversity of anuran amphibians.

## Figures and Tables

**Figure 1 animals-14-01406-f001:**
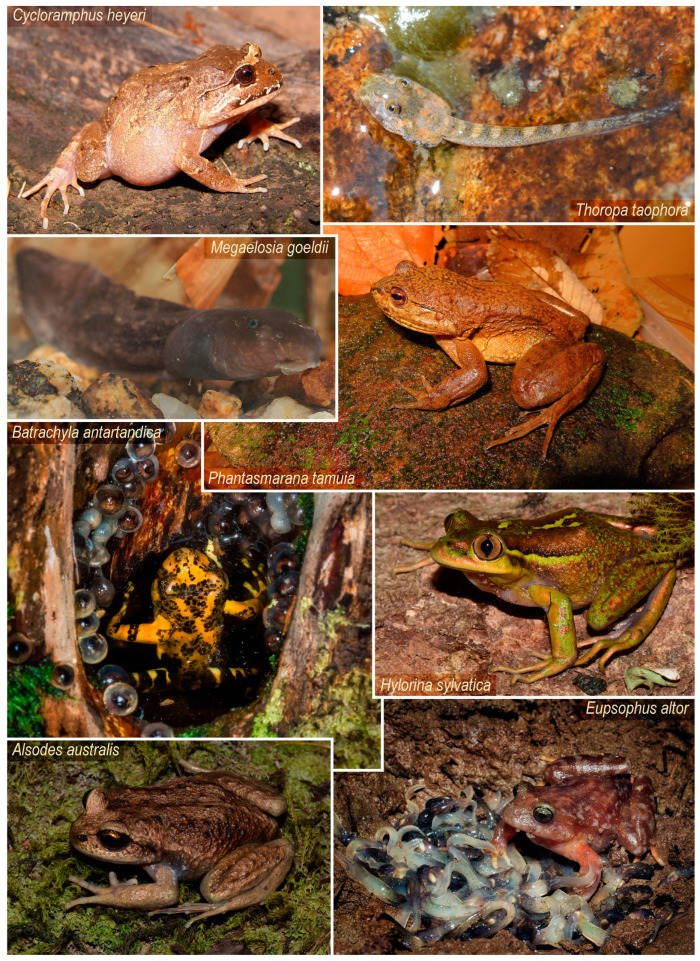
Larval and adult diversity in Neoaustrarana. Cycloramphidae: semifossorial adult of *Cycloramphus heyeri* and semiterrestrial tadpoles of *Thoropa taophora* (photos by F.F. Curcio and D. Baldo); Hylodidae: lotic benthic tadpole of *Megaelosia goeldii* and torrential adult of *Phantasmarana tamuia* (photos by F. Tubarão and L. Malagoli); Batrachylidae: terrestrial male of *Batrachyla antartandica* guarding eggs and pond-type tadpoles in a tree hole, and semiaquatic adult of *Hylorina sylvatica* (photos by F. Rabanal and D. Baldo); Alsodidae: adult specimen of *Alsodes australis* and male of *Eupsophus altor* guarding endotrophic tadpoles in a terrestrial nest (photos by D. Baldo and F. Rabanal).

**Figure 2 animals-14-01406-f002:**
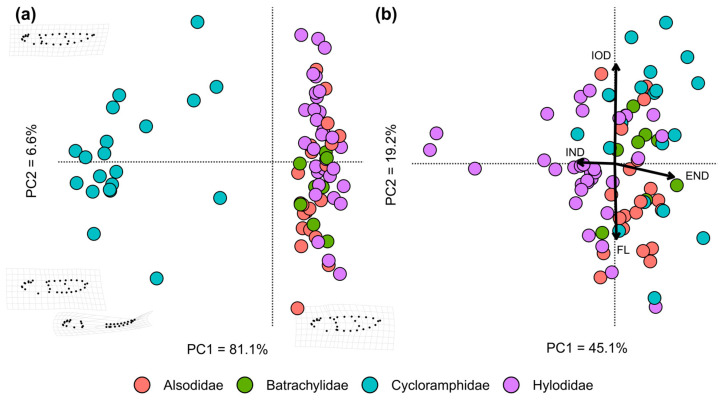
Larval (**a**) and adult (**b**) morphospaces for Neoaustrarana. Results of principal component analyses are shown, with species ordination on the two first principal components (PC). Deformation grids on larval morphospace illustrate shape change along axes, whereas arrows in the adult biplot are linear variables with higher correlations to axes. END: eye–nostril distance; IND: internarial distance; IOD: interorbital distance; FL: foot length.

**Figure 3 animals-14-01406-f003:**
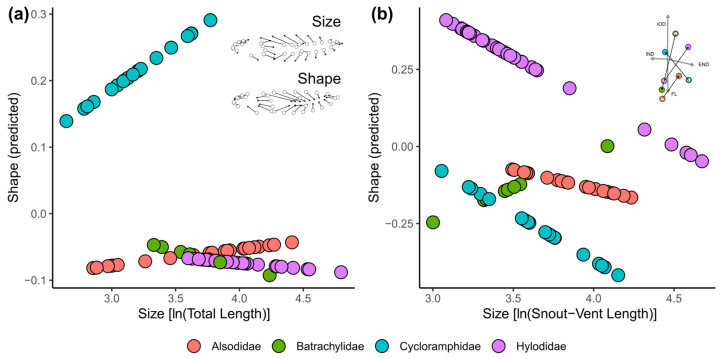
Allometric patterns in tadpoles (**a**) and adults (**b**) of Neoaustrarana. Shape is represented as the first principal component of a matrix of predicted shapes obtained from multivariate regression of shape on size. Deformation schemes depict shape changes along the axes, tending to higher body/tails in large larvae, and to flattened shapes with increasing PC1 score. In adults, allometric patterns can be interpreted from the biplot at the upper right, where the smallest (empty circles) and largest (solid circles) species are shown, and morphological changes are hinted by their ordination regarding relevant linear measurements. END: eye–nostril distance; IND: internarial distance; IOD: interorbital distance; FL: foot length.

**Figure 4 animals-14-01406-f004:**
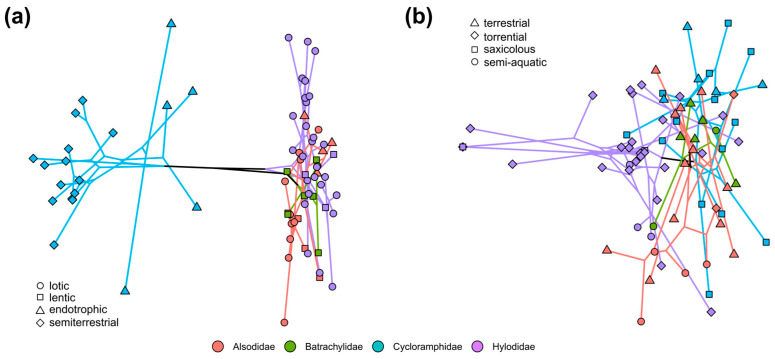
Larval (**a**) and adult (**b**) phylomorphospaces for Neoaustrarana. The phylogenetic hypothesis by Portik et al. (2023) is projected on morphospaces. Different symbols indicate larval and adult functional groups.

**Figure 5 animals-14-01406-f005:**
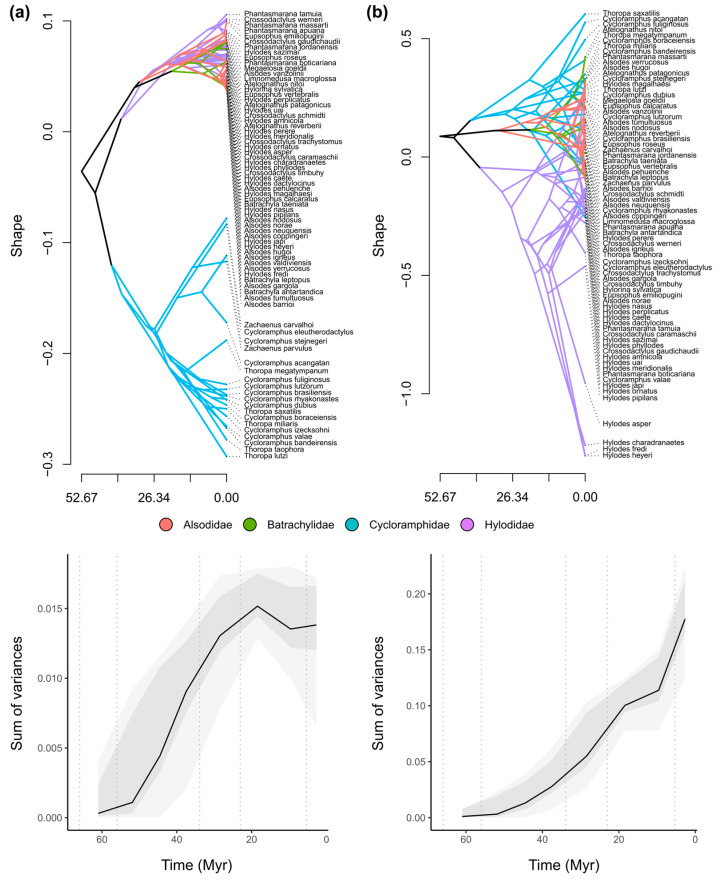
Evolutionary trends in shape of Neoaustrarana through time. Traitgram and morphological disparity (assessed from the sum of variances in ancestral estimation of PC1) of (**a**) larval and (**b**) adult shapes. Note the branch clustering as an indicator of shape similarity in the traitgrams, disregarding taxonomic structure.

**Figure 6 animals-14-01406-f006:**
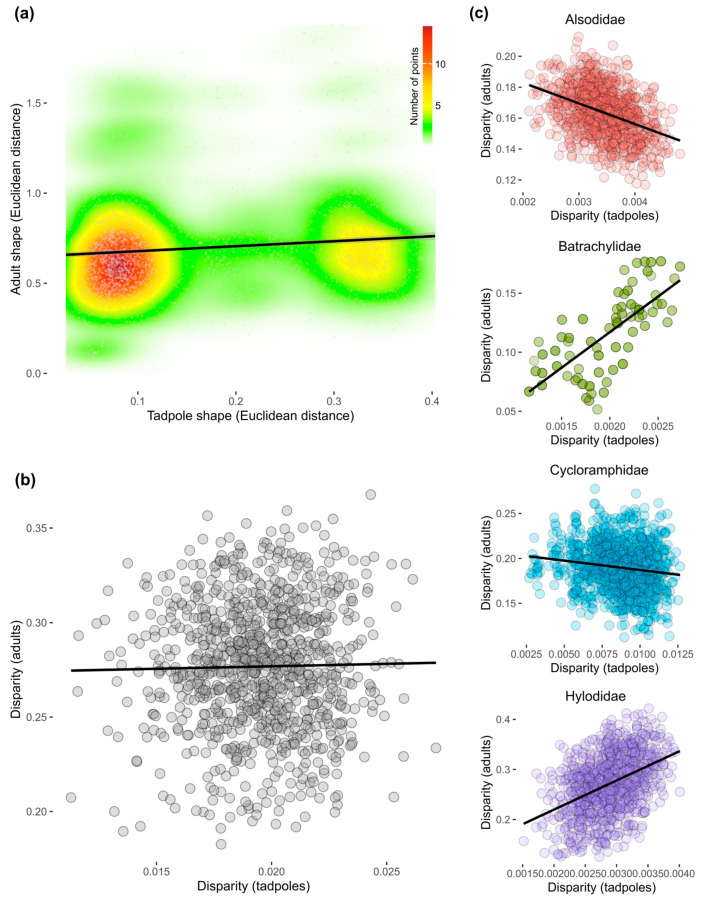
Morphological decoupling between larval and adult shapes in Neoaustrarana. (**a**) Kernel density map illustrating the correlation between tadpole and adult shape dissimilarity. Note that the low dissimilarity in the adult phase is not directly associated to some specific level of larval shape dissimilarity, indicating that adult shapes tend to be similar either when species with similar or different tadpoles are being considered. Correlation between morphological disparity of tadpole versus adult shapes, for the (**b**) entire dataset and (**c**) per family.

**Table 1 animals-14-01406-t001:** Shape and size relationship in larval and adult Neoaustrarana. Procrustes ANOVA of shape on size (larval total length and adult snout–vent length as natural logarithm-transformed values) by family.

	df	SS	MS	Rsq	F	Z	*p*-Value
Tadpoles: shape~ln(TL ^1^) × family							
ln(TL)	1	0.33499	0.33499	0.20377	849.852	43.726	<0.001
family	3	0.95021	0.31674	0.57798	803.536	74.096	<0.001
ln(TL):family	3	0.06318	0.02106	0.03843	53.427	46.174	<0.001
Residuals	75	0.29563	0.00394	0.17982			
Total	82	164.401					
Adults: shape~ln(SVL ^2^) × family							
ln(SVL)	1	1.6673	1.66729	0.07172	7.9340	3.6049	<0.001
family	3	4.6146	1.53821	0.19849	7.3197	5.1940	<0.001
ln(SVL):family	3	1.2055	0.40182	0.05185	1.9121	1.6838	0.051
Residuals	75	15.7610	0.21015	0.67794			
Total	82	23.2484					

^1^ TL: total length; ^2^ SVL: snout–vent length; df: degrees of freedom; SS: sum of squares; MS: mean squares; Rsq: R-squared; F: test statistics; Z: effect size.

**Table 2 animals-14-01406-t002:** Stayton’s C1–C4 measures of convergent evolution for larval and adult shapes in Neoaustrarana.

	C1		C2		C3		C4	
	Value	*p*	Value	*p*	Value	*p*	Value	*p*
Larval phylomorphospace								
Family								
Alsodidae	0.276	0.30	0.007	0.99	0.159	0.28	0.005	0.79
Batrachylidae	0.253	0.24	0.005	0.97	0.149	0.24	0.010	0.73
Cycloramphidae	0.422	**<0.01**	0.046	**<0.01**	0.235	**<0.01**	0.018	0.36
Hylodidae	0.487	**<0.01**	0.026	0.74	0.275	**<0.01**	0.008	0.57
Ecomorphological guild								
Lentic	0.500	**<0.01**	0.046	0.18	0.233	**0.02**	0.002	0.86
Lotic	0.582	**<0.01**	0.052	0.09	0.294	**<0.01**	0.005	0.52
Endotrophic	0.312	0.32	0.052	0.145	0.200	0.11	0.013	0.53
Semiterrestrial	0.685	**<0.01**	0.075	**<0.01**	0.381	**<0.01**	0.031	0.53
Adult phylomorphospace								
Family								
Alsodidae	0.296	0.15	0.050	0.50	0.162	0.20	0.021	0.27
Batrachylidae	0.216	0.49	0.045	0.33	0.144	0.31	0.009	0.82
Cycloramphidae	0.292	0.19	0.083	0.45	0.153	0.53	0.004	0.96
Hylodidae	0.380	**<0.01**	0.166	**<0.01**	0.194	**<0.01**	0.013	0.38
Ecomorphological guild								
Semi-aquatic	0.420	0.12	0.158	0.18	0.211	0.14	0.001	0.85
Torrential	0.357	**0.01**	0.168	**<0.01**	0.189	**0.02**	0.015	0.24
Saxicolous	0.303	0.25	0.126	0.12	0.164	0.30	0.004	0.84
Terrestrial	0.319	0.25	0.080	0.83	0.151	0.68	0.002	0.96

Bold values denote statistical significance at the *p* < 0.05 level.

## Data Availability

The data that support the findings of this study are available from the corresponding author upon reasonable request.

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
