# Peer review of "Shape Evolution in Two Acts: Morphological Diversity of Larval and Adult Neoaustraranan Frogs"

_animals, 2024, doi:10.3390/ani14101406_

Round 1

Reviewer 1 Report

Comments and Suggestions for Authors

The manuscript entitled “Shape Evolution in Two Acts: Morphological Diversity of Larval and Adult Neoaustraranan Frogs” evaluated the morphological diversity between larval and adult Neoaustraranan frogs.

The authors found a low correlation between larval and adult shapes, and results support the ADH driving phenotypic diversity in Neoaustrarana. This is a
complete article, but there are still some details that need to be improved.

Material and methods

1) All the data (photographs/drawings) were collected from the references, and only 1 or 2 data for each species. How to avoid the effect of data errors (from references) on the calculations and conclusions of this article.

2) During the developmental stages 26–41, the morphological structure of tadpoles is not a simple vector zoom. For example, the ratio of body/tail at early stage is higher than that at later stages. How does the author unify data from different developmental stages?

Discussion

Has the author considered the possibility that environmental pressures from predators, in conjunction with lotic and lentic habitats, could influence evolution of shape?

Supplementary Table S1

What is the abbreviation for NID? Need to clarify.

Author Response

Reviewer 1

1) All the data (photographs/drawings) were collected from the references, and only 1 or 2 data for each species. How to avoid the effect of data errors (from references) on the calculations and conclusions of this article.

2) During the developmental stages 26–41, the morphological structure of tadpoles is not a simple vector zoom. For example, the ratio of body/tail at early stage is higher than that at later stages. How does the author unify data from different developmental stages?

1 & 2. Thanks for these comments! Our goal was to explore differences at a high level of family and general functional groups. Thus, species sampling was performed to get a nice representation of that diversity. We assumed that intraspecific variations (e.g., development, drawing vs. photograph, individual variations) would be subsumed by variations at higher levels. Nevertheless, we are now including a Procrustes ANOVA to explore the effect of developmental stages in larval morphospace, which resulted non-significant as compared with other tested factors.

Has the author considered the possibility that environmental pressures from predators, in conjunction with lotic and lentic habitats, could influence evolution of shape?

Likewise, we are aware about the effect of several ecological factors (abiotic conditions, competition, predation) on tadpole phenotype, but our interest on shape variation at high taxonomic level lead us to assume that their impact could be disregarded in this context.

Reviewer 2 Report

Comments and Suggestions for Authors

Pleas see attached file

Comments on the Quality of English Language

Quality of english is high just some minor suggestions and corrections

Author Response

Reviewer 2

This paper was examining the phenotypic separation between juveniles and adults of a frog group with a diversity of lifestyles. It sought to identify the variability in form within a phylogenetic context. It was well written and a very interesting read. I had a few relatively minor concerns that probably just need greater clarity. The paper concerns itself with the Adaptive Decoupling Hypothesis. This would suggest that the evolution of the attributes of the tadpoles does not align with the evolution of characteristics of the adults. Given the adults and tadpoles live in different environments that is not that surprising in some ways. But irrespective the issue I have is the comparison of attributes. The characterisation of the tadpoles seemed mostly on shape analyses whereas the adults were on linear scale measurements. It was not clear how these two treatments would likely correlate if they were not decoupled? i.e. the null hypothesis here. And I think that is missing from the introduction. But also in the methods it is not clear how you would expect a unitless measure to be comparable with scaled measures. I know all of the information is placed in a dimensionless analysis. But there still has to be some underpinning biology that informs the characters being compared. So bearing that in mind then why were the same measurements of the adults not used for the tadpoles such as interocular measures and if leg length and tail lengths were comparable why were they not compared etc.

As we understand, it is possible to apply the conceptual frame of ADH to explore those characters that do have continuity between tadpoles and adults (e.g., body size) and those that don’t (most other features that involve characters, shapes, etc., that are present in tadpoles and not in adults due to metamorphosis). Since tadpoles are that different from adults, most of the studies we consulted deal with this second approach. Thus, the interpretation relays on decoupling of the pattern of similarities/differences among tadpoles and among adults. Some frog species can be very different among each other during larval stages, but once metamorphosed they are very similar, and the other way around. In fact, our results are opposed to those found in Australian clades. In this approach then, the way those characters are registered, we understand is minor: as long as we choose a suitable way to represent tadpole shape (i.e., geometric morphometrics in our case) and adult shape (i.e., size corrected linear measurements, which is also a proxy of shape), then the emphasis is put on decoupling at a morphospace level: tadpoles differ with cycloramphids being the most divergent, but when adults, cycloramphids are like other neoaustraranans and hylodids are the ones diverging.

Secondly the results suggest the tadpole characteristics fit better to the phylogeny than the adult characters. Thus are you suggesting that the divergence of the phylogenetic trees is due primarily to shifts in genes coding tadpole traits? If this is the case then maybe finding a simpler way of expressing this would be good at some point during the paper. Extending on this though is then asking if the traits you have chosen for the adult morphometrics are best aligned with their environmental needs? And this where I would suggest providing some more ecological/biological explanation of the chosen traits measured. For tadpoles the shape parameter is arguably more holistic then taking snout-vent length for the adults.

We have no clue about how development at a molecular level can be related with this. We can only guess that factors that determine the distinct shape of cycloramphid tadpoles are strong enough as to be evident in embryos that lost the semiterrestrial habit. As to adults, measurements considered are standard measurements included in species descriptions, so they have taxonomic value, no doubt, and could have some functional implication. Thus, if we understood properly, we see no risk of circularity taking into account some functionally/ecologically relevant measurements and finding functional groups supporting morphospace structure. In fact, in Sherratt et al. study, the same measurements are taken and their adult morphospace turned out better structured by phylogeny.

Line 116-118: The defining of landmarks. One of the issues with using a landmark approach is defining them. As this is a research paper they need to be defined so the work can be repeated. The supplementary image provides approx. location of the landmarks. Some of them are obvious as they are at noticeable junctures but for the ones on the tail, based on the image alone it would be hard to replicate those landmark locations. Can the author provide more guidance on the placement of these landmarks??

Sorry, this was likely a misunderstanding? Landmarks are defined in the captions of Fig. S1, and we understood the journal format required captions to be in the main text before the References? Now we included captions directly in the Figures, but we’ll wait for further instructions if we need to change that back.

I also went to the paper that defines the landmarks but they only have 20+ points and you have 34. Some of these are not landmarks as well according to that paper but semi-landmarks. So please provide full disclosure. Irrespective of whether the other paper has those details it would be better in my mind if you tabulised that information and have it accompany the figure you created.

Indeed, we used equidistant points along body outlines to register snout and tail shape (they are defined like that in Figure’s caption). We didn’t treat these points as semilandmarks though (i.e., we didn’t slide and adjust their position during Procrustes superimposition). This and other approaches to study of outlines are discussed by MacLeod (e.g., https://www.palass.org/publications/newsletter/palaeomath-101), which we mostly follow.

117-119: “Landmark were digitized by the same person, and configurations were digitally unbent to correct tail curvature” What does this mean? And how was it performed in an objective manner? What if the curvature is a defining feature i.e. curvy?

This is a standard tool in TPS to correct artificial curvature (actually developed after realizing that many specimens appeared curved in ordination plots, just for a post-mortem effect). First you define a “backbone line” so you choose which curvature to correct. We are sorry we were not clear enough, we added a brief comment and a citation in text.

Line 127: why were they grouped in functional groups prior to analyses should that not be a posthoc i.e. based on groupings of the objective data to see if it reflected the functional groups rather than the other way around?

Our first goal was to explore the shape morphospace. Functional groups, as well as phylogenetic structure (i.e., families and genera) were used first as “classifiers” to understand shape ordination.

Correct his sentence. And should this sentence be 'We assessed the alignment of species positionings on morphospaces using...'?

Done!

Are you able to provide a sample of tadpole and adult forms that illustrate the variations? It seems like the results are crying out for images of the actual tadpoles?

Thanks for this suggestion! We included a new Figure 1 showing a glimpse of larval and adult diversity in this group.

more in the caption to help interpret ‘whereby clustering of colours reflects greater relatedness’

Done!

Figure 5a: can you put a little more in the caption to help someone interpret the figure. It has hotspots and the hotspots align with what exactly. The result gives the impression there is a bimodal distribution rather than a cloud of dots like the other figures.

Done!

Paragraph starting line 430: The argument here if I am to follow is that there was variation in the prevailing conditions between times in which certain families arose. Ok that is fair enough but that explains variation between families but what about the decoupling within the species? As presumably the tadpoles and adults were subjected to the same conditions? In addition to that I think this paragraph is too long and seemingly tangential.

Our point here is that this landscape variation could be related with families from the Atlantic Forest having more pronounced differences between tadpoles and adults. The premise is that tadpoles and adults, since occupy different ecomorphological niches, may be subject to different environmental conditions. And this could be more significant in changing environments. Thus, while Patagonian tadpoles look comparatively similar, Brazilian cycloramphids and hylodis are radically different to each other. And the adult pattern is different from that of tadpoles but still involves a Brazilian family, this time hylodids, as the most divergent in morphospace occupation. We made the paragraph a bit shorter.

Round 2

Reviewer 1 Report

Comments and Suggestions for Authors

The added Procrustes ANOVA checked the variations in different developmental stages. I have no more questions. Thanks.